# Visual Storytelling with Question-Answer Plans

**Danyang Liu, Mirella Lapata, Frank Keller**
Institute for Language, Cognition and Computation
School of Informatics, University of Edinburgh
10 Crichton Street, Edinburgh EH8 9AB
danyang.liu@ed.ac.uk, {mlap, keller}@inf.ed.ac.uk

## Abstract

Visual storytelling aims to generate compelling narratives from image sequences. Existing models often focus on enhancing the representation of the image sequence, e.g., with external knowledge sources or advanced graph structures. Despite recent progress, the stories are often repetitive, illogical, and lacking in detail. To mitigate these issues, we present a novel framework which integrates visual representations with pretrained language models and planning. Our model translates the image sequence into a *visual prefix*, a sequence of continuous embeddings which language models can interpret. It also leverages a sequence of question-answer pairs as a *blueprint plan* for selecting salient visual concepts and determining how they should be assembled into a narrative. Automatic and human evaluation on the VIST benchmark (Huang et al., 2016) demonstrates that blueprint-based models generate stories that are more coherent, interesting, and natural compared to competitive baselines and state-of-the-art systems.

## 1 Introduction

Visual storytelling involves narrating an engaging and logically coherent story based on a sequence of images (see the example in Figure 1). The task lies at the intersection of natural language processing and computer vision and has recently attracted increasing interest from both communities (Wang et al., 2022; Hsu et al., 2021; Xu et al., 2021; Chen et al., 2021; Hsu et al., 2020; Wang et al., 2020; Huang et al., 2016). Visual storytelling differs from image captioning, which typically focuses on generating descriptive text, e.g., by identifying and depicting objects within an image. It requires a deeper understanding of how images and the events they illustrate relate to each other in order to create a convincing narrative.

Visual storytelling is commonly modeled as a two-stage process. The image sequence is first

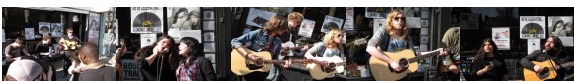

A small indie rock group performed for a music store in our city. The girls doing harmonies in the songs were phenomenal. The band had three guitarists. One of the guitarists resembled James Morrison. More independent bands played for the grand opening of the music store.

$Q_0$: What group performed for a music store in our city?
$A_0$: a small indie rock group
- - - - - - - - - - - - - - - - - - - - - - - -
$Q_1$: What kind of music did the girls do?
$A_1$: harmonies
$Q_2$: Who did the harmonies in the songs?
$A_2$: the girls
- - - - - - - - - - - - - - - - - - - - - - - -
$Q_3$: How many guitarists did the group have?
$A_3$: three guitarists
- - - - - - - - - - - - - - - - - - - - - - - -
$Q_4$: Who resembled James Morrison?
$A_4$: one of the guitarists
- - - - - - - - - - - - - - - - - - - - - - - -
$Q_5$: Who played for the grand opening of the music store?
$A_5$: more independent bands

Figure 1: Blueprint annotation for a visual story. Color-coded answers are extracted from the gold story. Questions are generated by feeding the answers and the gold story as context to a pretrained question generator.

encoded into a representation which typically includes image embeddings and detected objects. Subsequently, a decoder generates a story token by token based on the encoding of the image sequence. Recent work has mainly focused on enhancing the first stage of the generation process e.g., by leveraging external knowledge sources (Hsu et al., 2021; Chen et al., 2021; Hsu et al., 2020; Yang et al., 2019). Advanced representations for image sequences have also been explored, such as scene graphs (Hong et al., 2020) and story graphs (Hsu et al., 2021). Despite recent progress, these methods struggle to produce meaningful narratives, are prone to hallucination and repetition, often generate vague sentences, and have difficulty identifying salient visual concepts.

We attribute the lack of story quality to at least two reasons. Previous work on text-based genera-

tion has demonstrated that *planning* can improve story coherence, allowing to control the trajectory of events, the characters described and their actions (Yao et al., 2019; Xu et al., 2018; Rashkin et al., 2020; Goldfarb-Tarrant et al., 2020a; Fan et al., 2019; Yang et al., 2022). However, planning has not been considered in the context of visual storytelling, existing models adopt black-box architectures which are not particularly controllable or interpretable. Another limitation concerns the nature of current models which are essentially trained from scratch, and as a result have limited language modelling and generalization capabilities (they only see multimodal training samples; see top of Figure 1). Although pretrained language models (Raffel et al., 2020; Lewis et al., 2020; Brown et al., 2020) have been widely adopted for general-purpose story generation, their potential for visual storytelling remains unexplored.

In this work we propose an approach to visual storytelling which integrates pretrained language models with visual representations and incorporates an intermediate planning step before generating the full story. Our encoder translates the image sequence into a *visual prefix*, a sequence of continuous embeddings which language models can interpret. Following Narayan et al. (2022), we represent plans as a sequence of question-answer pairs, called *blueprints*, which serve as a proxy for content selection (i.e., what to say) and planning (i.e., in what order). Blueprints are loosely related to the Question-under-Discussion (QUD) theory of discourse (Larsson, 2002; Roberts, 2012; De Kuthy et al., 2020), which posits that text structure can be analyzed by identifying *implicit* questions raised and answered by subsequent spans of text. We augment visual storytelling training data with story blueprints (see Figure 1), which we obtain automatically thanks to state-of-the-art question generation technology.

We fine-tune pretrained language models to generate blueprints from image sequences *and* the stories based on them. We showcase two types of storytelling models, which vary in how the planning mechanism is implemented. A *top-down* model generates the blueprint first and then continues to generate the corresponding story in one go, whereas an *integrated* model interleaves planning with text generation rather than determining a plan in advance; generation is iteratively conditioned on the image input, the blueprint and the story gener-

ated so far. Experiments on the VIST benchmark (Huang et al., 2016) show that blueprint-based models generate more coherent, interesting, and human-like stories compared to the state of the art and large language models (LLMs) like GPT-3.5, according to automatic and human evaluation.

## 2 Related Work

**Visual Storytelling** Huang et al. (2016) introduced visual storytelling as a vehicle for developing AI tools with human-like understanding of grounded event structure and linguistic abilities that go beyond descriptive language. While earlier work (Gonzalez-Rico and Fuentes-Pineda, 2018; Kim et al., 2018) employed simple encoder-decoder architectures (using CNNs to extract visual features and RNNs to generate text), more recent methods (Xu et al., 2021; Chen et al., 2021; Hsu et al., 2020; Yang et al., 2019) leverage external resources (e.g., ConceptNet) as a way of instilling commonsense reasoning skills. Sometimes, scene graphs are also used to model relations between objects (Lu et al., 2016; Hong et al., 2020; Wang et al., 2020). To our knowledge, none of these approaches make use of plan-based decoding. Hsu et al. (2021) construct a graph representing the image sequence (based on training data and external resources) and identify the highest scoring path as the best storyline encapsulated therein. The storyline can be viewed as a form of planning, however, on the encoder side.

Most existing approaches (Xu et al., 2021; Hsu et al., 2020; Wang et al., 2020; Yang et al., 2019) train Transformer models from scratch, with the exception of Chen et al. (2021), who employ a vanilla BART model as a baseline without task-specific adaptation. In contrast, our work leverages the language modeling and generalization capabilities of pretrained language models for visual storytelling.

**Planning and Generation** In the domain of automatic story generation, planning has been effective at capturing the content and structure of stories. The generation process is often decomposed into two stages, namely planning an outline and then elaborating on it, e.g., by filling in specific details of a story. Plans have been represented as a sequence of event or phrase keywords (Yao et al., 2019; Xu et al., 2018; Rashkin et al., 2020), character actions (Liu et al., 2020), plot structures (Goldfarb-Tarrant et al., 2020a), and more elaborate descriptions including details about the setting

of the story, its characters, and main plot points (Yang et al., 2022).

The idea of having a separate planning stage has also been explored for other text generation tasks including summarization (Narayan et al., 2021; Liu and Chen, 2021) and data-to-text generation (Moryossef et al., 2019; Puduppully et al., 2022). Our work is closest to Narayan et al. (2022) who propose the use of question-answer pairs as intermediate plans for summarization. However, their approach is designed for descriptive text. Our work extends their framework to a multimodal setting, where the input consists of image sequences, and the output are narratives characterized by more abstract and figurative language.

## 3 Blueprint-based Visual Storytelling

Let $I$ represent a sequence of $k$ images, denoted as $\{v_1, v_2..., v_k\}$. Given this input, our goal is to generate a blueprint plan $B$ (i.e., an ordered set of question-answer pairs) and a story $S$ based on it. Most generation datasets do not include blueprint annotations, and visual storytelling is no exception. We first describe how we automatically obtain $\{I_i, B_i, S_i\}_{i=1}^N$ training data samples (Section 3.1), and then introduce our story generation models (Section 3.2).

### 3.1 Blueprint Annotation

Let $\{I_i, S_i\}_{i=1}^N$ denote a dataset consisting of pairs of image sequences and their corresponding stories. We automatically create blueprint $B_i$ based on story $S_i$ using state-of-the-art question generators (Romero, 2021; Raffel et al., 2020), coupled with a filtering procedure to remove repetitions and ill-formed questions.

The generation of question-answer pairs involves two steps, namely answer extraction and question generation. In the context of storytelling, capturing key events is crucial for a compelling narrative. While noun phrases and named entities are commonly recognized as significant content units in other tasks such as summarization (Narayan et al., 2022; Deutsch and Roth, 2023), verb phrases also play a vital role in conveying story dynamics, actions, and relationships (Trabasso et al., 1989; Eisenberg and Finlayson, 2017; Liu et al., 2020). Therefore, in addition to named entities and noun phrases, we also extract verb phrases as answer candidates using the spaCy library.

We then generate questions for answer candi-

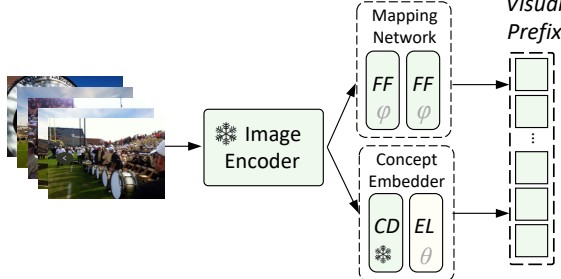

Figure 2: Visual prefix construction. The pretrained image encoder and concept detector are frozen. *FF* refers to a feed-forward layer, *CD* and *EL* denote a concept detector and embedding layer, respectively.

dates with a T5 model (Raffel et al., 2020; Romero, 2021) fine-tuned on the SQuAD reading comprehension dataset (Rajpurkar et al., 2018). The answer and the story are provided as context to predict the corresponding question. We decontextualize stories by replacing pronouns with their corresponding head mentions, using a state-of-the-art coreference resolution model (Dobrovolskii, 2021).

Question-answer pairs are subsequently filtered to eliminate noise which is unavoidable due to the automatic preprocessing steps mentioned above. We thus remove any question-answer pairs where the answer is already present in the question. We also employ a round-trip consistency check (Alberti et al., 2019) which discards questions if they yield answers different from those used to generate them.

### 3.2 Blueprint Models

Our approach leverages the generation capabilities of pre-trained sequence-to-sequence models. As our backbone model, we employ BART-base (Lewis et al., 2020) which has been fine-tuned for text generation. We adapt this model to our visual storytelling task in two ways. Aside from enabling the generation of blueprints, we convert the image sequence to a *visual prefix* which the pretrained language model can interpret. The pretrained language model is prompted with this prefix to generate the blueprint, and eventually the story.

**Visual Prefix Construction** Our model needs to grasp what the image sequence is about, e.g., the depicted objects, actions, and their associations. Drawing inspiration from recent advances in vision and language research (Mokady et al., 2021; Tsimpoukelli et al., 2021; Alayrac et al., 2022; Zhai et al., 2022; Liu et al., 2023; Huang et al., 2023), we translate the input sequence of images into a

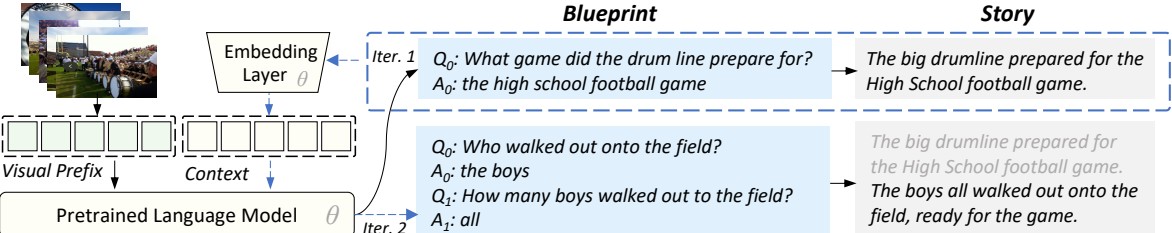

Figure 3: Iterative blueprint model: in the first iteration, the embedding layer uses *<START>* as context; in the second iteration, the context includes the blueprint and previously generated story (enclosed by blue dashed line). In contrast, the top-down model takes only the visual prefix as input and predicts a global blueprint and corresponding story in one go. For details on the visual prefix, see Figure 2.

sequence of continuous embeddings, aka a visual prefix (see Figure 2).

Following previous work (Wang et al., 2018; Xu et al., 2021; Chen et al., 2021), we use ResNet-152 (He et al., 2016) to extract visual features from images. We next employ a lightweight linear mapping network that consists of a series of feedforward layers, denoted as $F_\phi$, to map image features to $k$ visual clues:

$$p_1, \ldots, p_k = F_\phi \left( \text{ResNet} \left( v_1, v_2, \ldots, v_k \right) \right) \quad (1)$$

where $k$ is the image sequence length, and each visual clue $p_i$ has the same dimensionality as a token embedding of the pretrained language model.

To further instil world knowledge in our visual prefix, we employ a concept detector. The latter identifies specific objects within images, but also actions, scenes, and attributes. For each image $v_i$, we retain the $K$ concepts $\{c_i^1, c_i^2, ..., c_i^K\}$ with the highest confidence score:

$$c_i^1, \ldots, c_i^K = \text{Concept} \left( \text{ResNet} \left( v_i \right) \right) \quad (2)$$

Concepts for each image are then concatenated with a $\langle SEP \rangle$ token and serve as input to the embedding layer of the pretrained model. The visual clues and concept embeddings are concatenated to form the visual prefix $V$. The image encoder and the concept detector remain frozen during the training phase. Only the parameters in the mapping network $F_\phi$ are updated (see Figure 2).

**Top-down Planning** This model takes image sequence $I$ as input and generates blueprint $B$ and story $S$ in one go. More precisely, during decoding, the model first generates the blueprint, which then serves as a prompt guiding subsequent story generation. Our training objective maximizes the log-likelihood of the joint distribution:

$$\max_{\theta,\phi} \sum_{i=1}^{N} \log p_{\theta,\phi} \left( B_i, S_i \mid I_i \right) \quad (3)$$

where $(B, S)$ refers to the concatenation of blueprint $B$ and story $S$. $\theta$ represents the parameters of the pretrained language model, $\phi$ are the parameters of the mapping visual network $F_\phi$, and $N$ denotes the size of the dataset.

We introduce special tokens *Story:* and *Plan:* preceding the story and blueprint, respectively. In experiments, our blueprints consist of answer-question pairs $\{a_1, q_1, \ldots, a_m, q_m\}$ (rather than question-answer pairs). We place the answer before its question to encourage the model to zoom in on salient visual concepts depicted in the image sequence. This ordering is intuitive for our story-telling task: We first decide on what the story is about and then elaborate on key concepts. Incidentally, Narayan et al. (2022) also find that generating the answer before the question performs better for their summarization tasks. Finally, the model is trained with the standard maximum likelihood objective to generate the joint target.

**Iterative Planning** This model employs an incremental generation strategy to create the story. Rather than generating in one step a global blueprint and the story, planning and generation are interleaved. At each time step, the iterative model considers the image sequence *and* context from previous steps, including the blueprint and story generated so far. We gradually construct the blueprint and its corresponding story sentence-by-sentence; our planning is informed by generation and vice versa, which we argue should be mutually beneficial (they are conditioned on each other).

Let $S = \{s_1, s_2, \ldots, s_k\}$, denote a target story and $B = \{b_1, b_2, \ldots, b_k\}$ its blueprint where

| | |
|---|---|
| Length of image sequences | 5.0 |
| Number of sentences in the story | 5.0 |
| Number of tokens per story | 52.3 |
| Number of QA pairs per story | 11.1 |
| Number of tokens per QA pair | 10.3 |
| Number of tokens per story plus QA pair | 166.2 |

Table 1: VIST dataset Statistics (average values).

$s_i$ represents the $i$-th sentence in the story, and $b_i$ its associated blueprint. Each $b_i$ consists of answer-question pairs, denoted as $\{a_1^i, q_1^i, \ldots, a_{l(i)}^i, q_{l(i)}^i\}$, where $l_{(i)}$ is the number of pairs in the $i$-th blueprint. The training objective for the iterative model is defined as follows:

$$\max_{\theta,\phi} \sum_{j=1}^{N} \sum_{i=1}^{k} \log p_{\theta,\phi}\left(b_{i+1}, s_{i+1} | b_{1:i}, s_{1:i}, I_j\right) \quad (4)$$

where $b_{1:i}$ and $s_{1:i}$ refer to the blueprint and sentences generated so far, from step 1 to step $i$.

At each time step $i$, the encoder takes image sequence $I$ as input, and the decoder takes the context (i.e., blueprint and sentences generated so far $\{b_1, b_2, \ldots, b_i; s_1, s_2, \ldots, s_i\}$) as a prompt to predict the next blueprint $b_{i+1}$ and sentence $s_{i+1}$. Therefore, the iterative model is trained on samples $\{I, (b_{1:i}, s_{1:i}), b_{i+1}, s_{i+1}\}$. We prefix $(b_{1:i}, s_{1:i})$, $b_{i+1}$, and $s_{i+1}$ with *Context:*, *Plan:*, and *Next Sentence*, respectively. To handle the first time step, we introduce special token $\langle START \rangle$ as context to predict $b_1$ and $s_1$. We also use $\langle END \rangle$ to indicate the completion of an iteration (see Figure 3 for an illustration). It is important to note that $(b_{1:i}, s_{1:i})$ are masked out when computing the loss because they serve as prompts to the decoder. We want to avoid the model repeatedly predicting and overly optimizing the blueprints and sentences that appear at the beginning of the output.

# 4 Experimental Setting

## 4.1 Dataset

We performed experiments on the widely used VIST dataset (Huang et al., 2016), which contains 10,117 Flickr albums and 210,819 unique photos. Each training sample consists of $k = 5$ images and a corresponding story of $k = 5$ sentences. As described in Section 3.1, we augment each story with an automatically generated blueprint.

## 4.2 Implementation Details

Our models are built on top of BART-base (Lewis et al., 2020) and finetuned with a learning rate of 3e-5, batch size of 64, and warm-up ratio of 0.05. We select the best checkpoint on the validation set using a QA-based metric which quantifies the extent to which the output story follows its blueprint (see Section 4.4). During inference, we employ beam search (size 5). For our visual prefix, we employed the Clarifai Concept Detector which was trained on a dataset containing 9,098 concepts and 20 million images (multiple concepts are assigned to each image), and is integrated with the Inception V2 architecture (Szegedy et al., 2016) .

## 4.3 Comparison Systems

We compared our models against several baselines and state-of-the-art visual storytelling models. These included a vanilla BART-base model with the same encoder and visual prefix as ours but no planning (**VP-BART**; it generates the story directly in an autoregressive manner without the blueprint). **KG-Story** (Hsu et al., 2020) predicts a set of words representative of the image sequence, enriches them using external knowledge graphs, and generates stories based on the enriched word set. **PR-VIST** (Hsu et al., 2021) is a state-of-the-art model which constructs a graph representing the relations between elements in the image sequence, identifies the best storyline captured therein, and proceeds to generate a story based on it. The process of constructing the story graph can be viewed as a form of planning. Along similar lines, Chen et al. (2021) build a common sense knowledge graph capturing concepts in the image sequence, and use **MCSM**, a Maximal Clique Selection Module to identify which ones to write a story about. They use BART-large to generate the story based on selected concepts (and image features).

We also compared against an LLM that generates stories via prompting. We provide GPT-3.5 with a visual prefix, namely the concepts identified in the image sequence, and a prompt which explains how to create the blueprint and generate the story together with examples (in-context learning). Details on the prompt can be found in Appendix A.

## 4.4 Automatic Evaluation

We evaluated our stories using BLEU, ROUGE, METEOR, and CIDER, mainly to compare to previous work. Several studies (Hsu et al., 2022, 2021,

2020; Hu et al., 2020; Yang et al., 2019; Modi and Parde, 2019) have demonstrated the inadequacy of lexical matching metrics: they correlate poorly with human judgments, and not do effectively measure the semantic similarity to human-written stories or the lexical richness of the generated stories.

We further employ *story-specific* metrics to assess story quality aspects such as diversity, fluency, naturalness, and grounding. Specifically, we use two types of trigram repetition metrics (Yao et al., 2019; Goldfarb-Tarrant et al., 2020b). *Intra-story repetition* is a fluency metric, it measures the proportion of trigrams repeated *within* a story. *Inter-story repetition* examines trigram repetition *across* stories. This metric evaluates diversity, high intra-story repetition suggests that the model tends to generate the same story even when conditioned on different image sequences. We also use MAUVE (Pillutla et al., 2021) to measure the *naturalness* of the generated stories. MAUVE is a recently introduced automatic metric for open-ended generation which has high correlation with human judgements. It computes the similarity of the distribution of human-written text and machine-generated text.

To quantify the extent to which the generated story is *grounded*, i.e., whether it accurately represents the content of the image sequence, we measure *concept precision* and *recall*. Precision measures the number of words in the generated story that align with the detected concept set, while recall assesses the number of words in the detected concept set that are present in the generated story.

Finally, for our own models we also evaluate whether the generated stories are *faithful* to their blueprint. Drawing inspiration from recent studies on summary evaluation (Deutsch et al., 2021; Fabbri et al., 2022), we measure how well the generated story answers questions from the predicted blueprint. We utilize a RoBERTa-based (Liu et al., 2019) QA model finetuned on the SQuAD dataset.

## 5 Results

Our results are summarized in Table 2. The first block presents the performance of state-of-the-art storytelling systems. The second block presents variants of our approach: a vanilla BART model, enhanced with a visual prefix (VP), and two blueprint models which vary in the way plans are generated, i.e., in a top-down fashion or iteratively. The third block contains GPT-3.5 models with (+BP) and without blueprints.

**Pretrained Language Models Produce Better Stories** We observe that models based on pretrained language models (i.e., our models and MCSM, outperform models trained from scratch (i.e., KG-Story and PR-VIST) in terms of trigram-repetition scores and MAUVE. This indicates that we can maintain strong language modeling capabilities while enabling pretrained language models to process visual signals effectively.

**The Visual Prefix is an Effective Interface between Image and Text** MCSM is the only existing model that utilizes a pretrained language model for visual storytelling. However, our baseline model (VP-BART) demonstrates superior performance in most story-specific metrics. Remarkably, this is achieved using a smaller pretrained model (BART-base, 140M parameters); MCSM is built on top of BART-large (400M parameters). This highlights the effectiveness of our visual prefix, indicating it successfully translates the image sequence into a space that BART can understand.

**Blueprint Models are Most Grounded** Our models outperform comparison systems in terms of concept grounding. This confirms that an intermediate planning step allows the model to effectively select salient concepts based on the visual prefix. The top-down model in particular achieves the highest concept grounding recall, it stays close to the image sequence, accurately describing the information conveyed therein. The higher lexical matching scores further support this observation. The iterative blueprint model achieves the best concept grounding precision (excluding GPT-3.5 models) which in turn suggests that the stories generated by this model exhibit a stronger grounding to the images with fewer hallucinations.

**The Iterative Model Generates Most Natural and Faithful Stories** Despite not achieving the highest scores in lexical matching metrics, the iterative blueprint model stands out in terms of MAUVE evaluation. Compared to other models, it generates more natural stories, closer to those written by humans. This finding suggests that humans might employ a similar iterative planning strategy, at least for the short stories considered here; they construct a narrative gradually rather than a global plan which they subsequently convert into a story.

With regard to faithfulness, we observe that both blueprint models achieve scores higher than 40%, indicating effective translation of blueprints into

| Model | Repetition (↓) | | Grounding (↑) | | MAUVE (↑) | Faithful (↑) | N-gram-based Metrics (↑) | | | |
|---|---|---|---|---|---|---|---|---|---|---|
| | Intra | Inter | Precis. | Recall | | | B-4 | RLSum | METEOR | CIDER |
| KG-Story | 1.03 | 88.72 | 4.55 | 3.46 | 3.86 | — | 9.8 | 27.3 | 32.3 | **7.9** |
| PR-VIST | 1.19 | 83.80 | 3.76 | 3.28 | 2.31 | — | 7.5 | 26.1 | 31.4 | 7.6 |
| MCSM | 2.85 | 77.48 | 5.12 | 5.89 | 11.01 | — | 8.1 | 27.7 | 31.4 | 7.6 |
| VP-BART | 0.22 | 83.70 | 4.31 | 3.23 | 11.31 | — | 8.6 | 26.6 | 31.0 | 6.8 |
| + BP (top-down) | **0.08** | 81.51 | 5.17 | **11.56** | 8.32 | 44.73 | **9.9** | **28.5** | **33.6** | 7.2 |
| + BP (iterative) | 0.29 | **72.70** | **5.22** | 3.59 | **28.25** | **51.66** | 7.0 | 26.1 | 30.3 | 5.5 |
| + BP (gold) | 0.12 | 18.61 | 6.81 | 2.97 | 52.24 | — | 29.4 | 52.0 | 58.4 | 36.3 |
| GPT-3.5 | 0.47 | 40.61 | 10.80 | 7.90 | 2.30 | — | 5.0 | 24.4 | 27.3 | 1.9 |
| GPT-3.5 + BP | 1.52 | 31.19 | 14.70 | 10.30 | 2.10 | 34.56 | 4.2 | 23.3 | 25.1 | 2.3 |

Table 2: Automatic evaluation results. We report intra- and inter-story trigram Repetition (lower is better), precision and recall for concept grounding, MAUVE, Faithfulness, and a suite of commonly used metrics which rely on lexical similarity between system stories and references. Best results are highlighted in bold font.

stories. Notably, the iterative model performs best in terms of faithfulness, which suggests that translating the entire global blueprint into a story is more challenging, whereas breaking down planning into individual steps is more effective. To get an idea of the upper bound performance for blueprint models, we ran the top-down model with silver standard blueprints extracted from the human-written stories (see row +BP (gold) in Table 2). As can be seen, the MAUVE score jumps to 52.24, edging closer to human-written stories (their MAUVE score is 69.6). This further supports our hypothesis that our model successfully leverages the blueprints and retains the information captured in them.

**GPT-3.5 Struggles with Blueprints** We also compared our approach to GPT-3.5, which we adapted to our task with in-context learning. A GPT-3.5 model enhanced with blueprints performs well at concept grounding, i.e., it generates stories which talk about what is depicted in the image. However, these stories are neither human-like (see the very low MAUVE score) nor faithful to the intermediate blueprints (in fact they are 10% less faithful compared to our iterative model). This suggests that GPT-3.5 tends to ignore the plan, despite being explicitly prompted with blueprints.

## 6 Human Evaluation

We conducted a judgment elicitation study to further evaluate the stories generated by our models. Specifically, we compared the best performing blueprint model (iterative) and three other systems: (a) PR-VIST, which represents the current planning-based state of the art; (b) VP-BART, our proposed model without blueprints; and (c) ground truth stories written by humans. Raters were shown an image sequence, alongside two stories and were

asked to provide pairwise references along the following dimensions: *Relevance*, *Fluency*, *Coherence*, *Interestingness*, and *Overall*. The full instructions are given in Appendix A. We recruited nine native English speakers and elicited preferences for 100 stories (three judgments per story).

Our human evaluation results are summarized in Table 3. The iterative blueprint model outperforms PR-VIST across metrics. Our participants perceive VP-BART stories as marginally more fluent and coherent compared to those created by the iterative model (even though they prefer iterative stories overall). This discrepancy is likely due to the generation process introduced by the iterative model which requires the decoder to produce a mix of questions, answers, and corresponding sentences, deviating from the traditional BART pretraining pattern. This added complexity might result in minor grammatical errors and pose challenges for coherence, given that story generation is broken down into separate steps instead of being a continuous process. Nonetheless, the coherence scores are fairly close.

The blueprint model excels in terms of interestingness and grounding, indicating its effectiveness in creating engaging and memorable stories. Our model's superior grounding performance aligns with our hypothesis that blueprints serve not only as a planning strategy but also as a visual concept selector. This is due to the way blueprints are structured (as answer-question pairs), which explicitly forces the model to first identify salient visual concepts and then generate questions based on them.

Figure 4 shows example stories created by the models used in our human evaluation study and GPT-3.5+BP. The story generated by the iterative model is coherent, rich in detail, and fluent. VP-

| Choices(%) | Iterative vs. PR-VIST | | | Iterative vs. VP-BART | | | Iterative vs. Human | | |
|---|---|---|---|---|---|---|---|---|---|
| | Win | Lose | Tie | Win | Lose | Tie | Win | Lose | Tie |
| Fluency | **47.0** | 37.6 | 15.4 | 40.5 | **42.9** | 16.6 | 10.9 | **84.3** | 4.8 |
| Coherence | **56.0** | 24.8 | 19.2 | 41.2 | **41.6** | 17.2 | 15.2 | **75.7** | 9.1 |
| Interestingness | **70.5** | 19.2 | 10.3 | **55.5** | 38.1 | 6.4 | 23.0 | **70.0** | 7.0 |
| Grounding | **50.9** | 40.6 | 8.5 | **45.1** | 41.7 | 13.2 | 7.8 | **79.6** | 12.6 |
| Overall | **58.1** | 25.6 | 16.2 | **42.6** | 41.5 | 15.9 | 12.6 | **80.0** | 7.4 |

Table 3: Human evaluation results. Raters provide pairwise story preferences in terms of fluency, coherence, interestingness, grounding, and overall. VP-BART is a BART model enhanced with a visual prefix (VP) but no planning; Iterative is our best blueprint model and PR-VIST is a state-of-the-art visual storytelling model. We report the percentage of times the Iterative model Wins, Loses or is in a Tie with a comparison system. Unless underlined, differences between systems are statistically significant ($p < 0.05$; using the Wilcoxon signed-rank test).

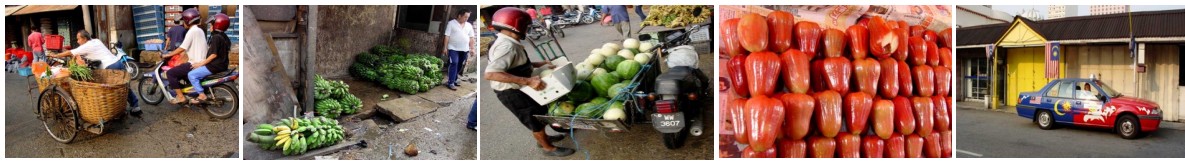

**PR-VIST**: I bought a cart to the market with some food. It looked great in the market. There were so many of their enthusiasm. They even had a sale with their bumper. The market was sold out of fresh vegetables. the [location] car is very nice and beautiful.

- - - - - - - - - - - - - - - - - - - - - - - - - - - - - - - - - - - - - - - - - - - - - - - - - - - - - - - - - - - - - - -

**VP-BART**: I went to the market yesterday. There were many different kinds of fruits there. I bought a lot of them. They were very expensive. Afterward I went back home.

- - - - - - - - - - - - - - - - - - - - - - - - - - - - - - - - - - - - - - - - - - - - - - - - - - - - - - - - - - - - - - -

**Iterative**: Today we decided to take a small shopping trip to the Market. The market had so many wonderful things to choose from that we looked at and bought so many of them. There was an array of different fruits that I could hardly resist buying. There was also a huge array of various types of pepper. Finally the day had come to an end and we piled in the taxi back to head home.

- - - - - - - - - - - - - - - - - - - - - - - - - - - - - - - - - - - - - - - - - - - - - - - - - - - - - - - - - - - - - - -

**GPT-3.5 + BP:** On Saturday morning, I visited the local market. It had a lot of fruits and vegetables being sold. The atmosphere was friendly and buzzing with many vendors ready to help you. The fresh produce was the best thing about the market. You can always find friendly vendors that sell the most delicious produce. The key to finding the best bargains at the market is bargaining. The market is always a great place to shop for fresh fruits and vegetables.

- - - - - - - - - - - - - - - - - - - - - - - - - - - - - - - - - - - - - - - - - - - - - - - - - - - - - - - - - - - - - - -

**Human:** Shoppers arrived early for the market. There was a variety of goods sold. Some carried away large bundles. Every item was fresh and colorful. The spot was a favorite among produce shoppers.

Figure 4: Examples of system output and human-written story for an image sequence.

BART generates a grounded and accurate story without hallucination and semantic errors. However, it is a relatively plain narrative, offering limited detail about the market or the experience of the characters. Compared to GPT-3.5+BP, the iterative model's story follows the image sequence more closely, mentioning details like *an array of different fruits* and *various types of pepper*, which significantly enhances storytelling.

## 7 Controllable Generation

In this section we showcase how the blueprint plan allows us to control the content and length of model output without additional mechanisms or training.

For example, in cases where the generated story contains entities which do not appear in the image sequence, it is possible to refine the story generation process, mitigating hallucinations. Specif-

ically, we apply a filtering step which removes non-grounded entities (and corresponding QA pair) from the blueprint before generating the story. We consider as *non-grounded* any blueprint entity which is not included in the output of the concept detector (see Section 3.2).

Figure 5 shows how this refinement approach can be used to adjust the model's output. In the first example, we observe that the story generated with a refined blueprint effectively avoids hallucinations (highlighted in blue) and is overall more faithful. However, it is important to note that imagination plays a crucial role in crafting an engaging story, especially when the image sequence provides limited information. Therefore, employing the refinement method may result in shorter and less detailed stories, as illustrated in the second example. While the refined blueprint successfully eliminates all hal-

**Faithful Story Generation**

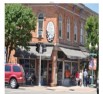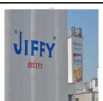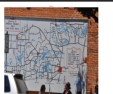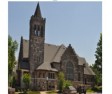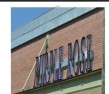

**Iterative:** We took a trip to Brazil. We went to the big City where we stay in, and took a lots of pictures. When we arrived, there were oompa pompa loompa hompa, and the people were walking around. We were at the altar, and it was standing. We ended the day getting hungry, so we decided to eat at a little elegant restaurant.

- - - - - - - - - - - - - - - - - - - - - - - - - - -

**Iterative + Grounded-Ent:** We were so excited to see where our vacation was going. We arrived at the hotel and it was quite busy. We couldn't believe how crowded it was getting. Some of the architecture was very cool. This store was our favorite since it was close to the city center.

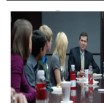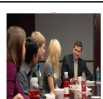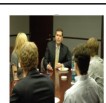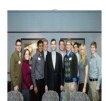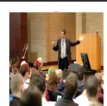

**Iterative:** I went to my colleagues the other day. My boss was the one that invited me to attend the meeting. He talked about how to improve the company for the future. After the presentation, I looked and listened to my coworkers on how the company would be doing. My boss was pleased that I invited them to talk for hours. We talked for hours and we were very pleased when my boss asked.

- - - - - - - - - - - - - - - - - - - - - - - - - - -

**Iterative + Grounded-Ent:** There was a question and answer session. Everyone was listening. The speaker was very interesting. Everyone was taking notes. The speaker talked very long.

Figure 5: Comparison of stories generated by original blueprint and refined blueprint models. Hallucinated words are highlighted in blue.

lucinated entities, the resulting story appears plain and lacks depth. Our blueprint method seems to strike the right balance between accurate and captivating story generation, prioritizing faithfulness to the image sequence and creativity in storytelling.

Most visual storytelling systems generate 5-sentence stories, following the predefined story structure of the VIST dataset (Huang et al., 2016). Nevertheless, our iterative blueprint model can flexibly modulate the length of the story by controlling the number of iterative steps, thereby overcoming the conventional sentence limitation. Figure 6 presents stories generated by this model with a maximum of 10 iterations. Despite the increased length, the stories maintain coherence and are engaging.

## 8 Conclusion

In this work, we have introduced a novel approach to visual storytelling which integrates visual representations with pretrained language models and a blueprint-based planning method for story gen-

**Long Story Generation**

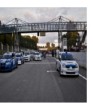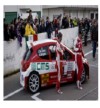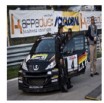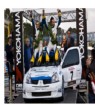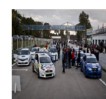

We took a car out onto the street that day. Here we are looking at the car that the winner of the race was in. The car with the winner in blue was very close throughout the whole race. It was neck and neck as the one in blue held the lead throughout the entire race. Finally, we crossed the lap and the winner pulled in for first place at the landing line. We were very close to winning the race and finishing in the final place line. It was a very very, very close finish!

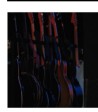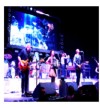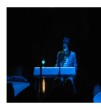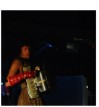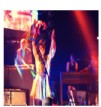

Last night, the band played an awesome concert. The guitarist had an incredible performance. The keyboard player was incredible as well. The guitarist was superb. He played his guitar very well. The next song he sang was very good. They had such a good performance, and I'm glad they went. The audience loved them so much. The night was over when the band wrote and put together their music. It was a fantastic way to finish the night.

Figure 6: Stories generated within 10 iterations.

eration. Blueprint models leverage a sequence of question-answer pairs as intermediate plans, enabling better selection of salient concepts from the image sequence and guiding the construction of the final narrative. Specifically, we have showcased two model variants: a top-down model which relies on a global plan, and an iterative model, which interleaves planning with sentence generation. Our experiments have shown that blueprint models excel in concept grounding and their ability to create human-like stories. Additionally, they are controllable: Blueprints can be made shorter or longer and their details can be refined (e.g., by emphasising specific entities or characters), thus enabling human-in-the-loop and personalized storytelling. We showcase examples of controllability in Section 7. In the future, we would like to explore visual storytelling with grounded characters and entities, as well as tackle the generation of more complex narratives, such as long-form stories.

**Acknowledgments** The authors gratefully acknowledge the support of the UK Engineering and Physical Sciences Research Council (grant EP/W002876/1). Liu was supported by the UKRI Centre for Doctoral Training in Natural Language Processing, funded by the UKRI (grant EP/S022481/1) and the University of Edinburgh.

## Limitations

While our proposed model demonstrates effective story generation, it has certain limitations. Firstly, the grounding relation between the visual concepts and the corresponding text may not always be clear, leading to potential ambiguity in the generated stories. Furthermore, the model can sometimes suffer from hallucinations due to falsely detected visual concepts.

It is worth noting that our model was built on top of BART-base (Lewis et al., 2020). It would be beneficial to investigate the performance of larger models, as they could potentially enhance the quality of the planning component and overall storytelling capability.

## Ethics Statement

**Large Language Models** This paper uses large pretrained language models, which have been shown to be subject to a variety of biases, to occasionally generate toxic language, and to hallucinate content. Model output used for the human evaluation study (Section 6) was screened by the authors for harmful content.

**Experimental Participants** The departmental ethics panel judged our human evaluation study to be exempt from ethical approval, as all participants were employees of the University of X, and as such were protected by employment law. Participants were paid at the standard hourly rate for tutors and demonstrators at the university.

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

## A  GPT-3.5 Experimental Setting

We designed the following prompts for GPT-3.5:

**Prompt (w/o Blueprint):**  *I want you to act as a visual storyteller by creating a story based on a given sequence of images. For each image, I'll provide the key concepts. The concepts from different images will be separated by <SEP>. \n Concepts: ..., \n Story: ..., ..., \n Concepts: ..., \n Story:*

**Prompt (w/ Blueprint):**  *I'd like you to act as a visual storyteller by creating a story based on a given sequence of images. For each image, I'll provide the key concepts separated by <SEP>. Your task is to first generate a series of question-answer pairs for each image as part of the planning process, and then use these pairs to create the final story. \n Concepts: ..., \n Plan: ..., \n Story: ..., ..., \n Concepts: ..., \n Plan: ..., \n Story:*

In-context learning examples in the prompts are not shown for brevity. To make the best of in-context learning, we employed max-shot learning, while adhering to the token limit of 4,096.

## B  Human Evaluation Study

As mentioned in Section 6 we conducted a judgment elicitation study to further evaluate the stories generated by our models. Specifically, we compared the best performing blueprint model (iterative) and three other systems: (a) PR-VIST, which represents the current planning-based state of the art; (b) VP-BART, our proposed model without blueprints; and (c) ground truth stories written by humans. Raters were shown an image sequence, alongside two stories and were asked to provide pairwise preferences along various dimensions of story quality. We describe below our evaluation procedure and reproduce the instructions given to our raters.

### B.1  Evaluation Procedure

We initially conducted a pilot study, based on which we devised our instructions. Subsequently, 100 image sequences were randomly selected from the test set, leading to a total of 300 pairwise comparisons. We employed the expertise of 9 native speakers who triple-annotated the 300 pairwise comparisons, resulting in a total of 900 judgments.

### B.2  Experimental Instructions

Human raters were asked to compare and evaluate stories generated by different systems using pair-wise judgments. The evaluation focused on the following dimensions of story quality: *Relevance*, *Fluency*, *Coherence*, *Interestingness*, and an *Overall* judgment. We provide their definitions below.

**Relevance**   Relevance captures whether the sentences in the stories relate to the input images. For each sentence, if the sentence accurately describes the content of a specific image (i.e., not imagining something that does not exist in the image), it will be marked as relevance. Otherwise, if the sentence does not correspond to an image or is too vague, it will be not marked as relevance.

**Fluency**   Fluency evaluates the grammatical correctness of the text. A story is fluent if it has few or no grammatical errors and is easy to understand.

**Coherence**   Coherence assesses whether the story makes sense. A coherent story flows well, the sentences are related, and logically connected. In contrast, an incoherent story would be more or less incomprehensible, without any logical connection between its sentences.

**Interestingness**   This evaluates whether the story contains unique, possibly unexpected elements. For example, a memorable storyline. Below are examples of a dull story and an interesting story.

**Overall**   Taking all the aforementioned criteria into consideration, the annotators selected their preferred story for the given set of five images.

### B.3  Annotation Interface

We designed an annotation interface using Python Flask. Figure 7 shows a screenshot of the interface.

# Annotation Interface

Progress: 5/100

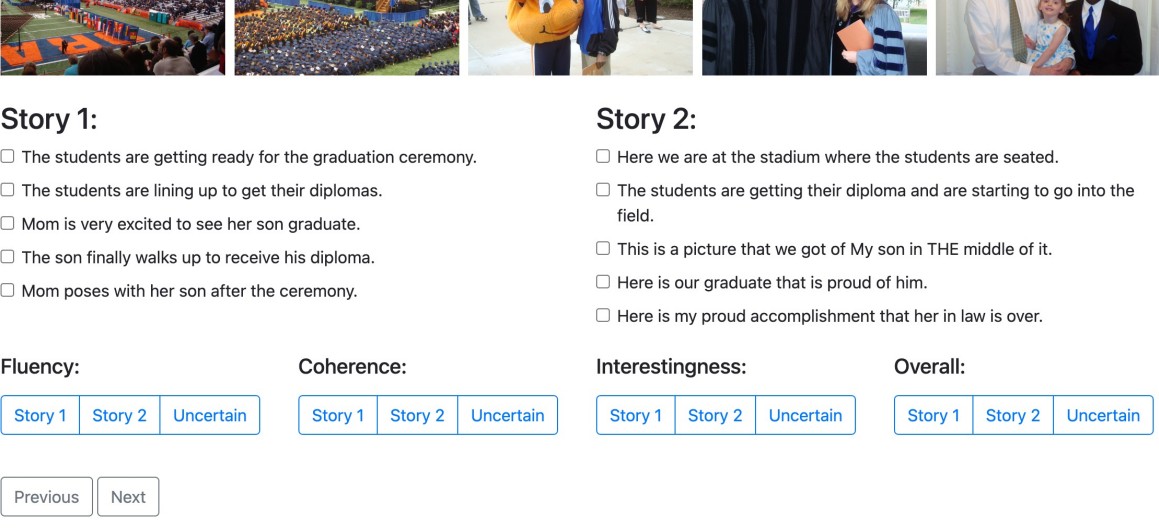

## Story 1:

☐ The students are getting ready for the graduation ceremony.

☐ The students are lining up to get their diplomas.

☐ Mom is very excited to see her son graduate.

☐ The son finally walks up to receive his diploma.

☐ Mom poses with her son after the ceremony.

## Story 2:

☐ Here we are at the stadium where the students are seated.

☐ The students are getting their diploma and are starting to go into the field.

☐ This is a picture that we got of My son in THE middle of it.

☐ Here is our graduate that is proud of him.

☐ Here is my proud accomplishment that her in law is over.

**Fluency:**

| Story 1 | Story 2 | Uncertain |

**Coherence:**

| Story 1 | Story 2 | Uncertain |

**Interestingness:**

| Story 1 | Story 2 | Uncertain |

**Overall:**

| Story 1 | Story 2 | Uncertain |

Previous  Next

Figure 7: Screenshot of the annotation interface.