# OpenReview forum: "Visual Storytelling with Question-Answer Plans"
_EMNLP/2023/Conference — EMNLP 2023 Findings_

### Official Review · Reviewer_B4N7 · 2023-07-23

**Soundness:** 4

**Excitement:**

3: Ambivalent: It has merits (e.g., it reports state-of-the-art results, the idea is nice), but there are key weaknesses (e.g., it describes incremental work), and it can significantly benefit from another round of revision. However, I won't object to accepting it if my co-reviewers champion it.

**Missing References:**

[1] Adversarial Reward Learning for Visual Storytelling. Xin Wang, Wenhu Chen, Yuan-Fang Wang, and William Yang Wang. In Proceedings of ACL, 2018.

[2] GLAC Net: GLocal Attention Cascading Networks for multi-image cued story generation. Taehyeong Kim, Min-Oh Heo, Seonil Son, Kyoung- Wha Park, and Byoung-Tak Zhang. In Proceedings of NAACL StoryNLP Workshop, 2018.

**Paper Topic And Main Contributions:**

This paper presents a method that leverages the blueprint (i.e., a set of question-answer pairs regarding the given image) to generate coherent and interesting narratives for the task of visual storytelling.

With a carefully designed process, the authors automatically collect the blueprint annotation given images and corresponding textual stories. The pre-trained language model (PLM) then generates the stories given visual prefixes and the blueprints. The parameters of the PLM remain frozen, and some parameters in the visual prefix construction are updated. This paper introduces two variants to generate visual stories: top-down planning and iterative planning.

Experimental results show that the proposed method outperforms some existing studies along with the ablative model. Furthermore, the authors perform human evaluation to demonstrate the effectiveness of the generated blueprints.


**Reasons To Accept:**

- This paper proposes the plan-based approach for visual storytelling, which has not been considered in existing studies.

- The authors perform human evaluation to identify the quality of the generated visual stories.

- Overall, the paper is well-written and easy to follow.


**Reasons To Reject:**

- The true impact of the generated blueprints seems questionable. In Table 2, the performance gap between VP-BART (w/o blueprints) and the proposed method (w/ generated blueprints) is much smaller than the performance gap between the proposed method and the model with gold blueprints, described as the upper bound performance. It indicates that the generated (i.e., silver) blueprints are noisy. Some techniques are required to reduce the noise in the generated blueprints.

- The current manuscript does NOT compare the proposed method with some strong and pioneering studies [1,2] in the experiments. In my current understanding, the authors exclude them because the N-gram-based performance of the proposed method lags behind that of those studies. It would be much more convincing if the authors compare the proposed method with them. The source code of both studies is open-sourced.


**Reproducibility:**

4: Could mostly reproduce the results, but there may be some variation because of sample variance or minor variations in their interpretation of the protocol or method.

**Reviewer Confidence:**

4: Quite sure. I tried to check the important points carefully. It's unlikely, though conceivable, that I missed something that should affect my ratings.

**Typos Grammar Style And Presentation Improvements:**

- L179: the index of the image should start from 1 if $k$ images are given

- L256: the index of the image should start from 1

- L264: the index of the concepts should start from 1 because the authors said that there are $K$ concepts.

- L316-317: the index should also start from 1

- L324-325: the notation that describes a set of elements from index 0 to index i is commonly represented as “0:i” rather than “0,i”

---

> ### Author Rebuttal · Authors · 2023-08-28
>
> > Q1: The true impact of the generated blueprints seems questionable. In Table 2, the performance gap between VP-BART (w/o blueprints) and the proposed method (w/ generated blueprints) is much smaller than the performance gap between the proposed method and the model with gold blueprints, described as the upper bound performance. It indicates that the generated (i.e., silver) blueprints are noisy. Some techniques are required to reduce the noise in the generated blueprints.
>
> Response: Thank you for your feedback!
>
> You are referring to the difference between VP-BART (w/o blueprints) and the proposed method (w/ generated blueprints)  when using n-gram based metrics. Table 2 reports additional metrics which reveal bigger differences between the two models. Using iterative blueprints results in an improvement of 16.94 MAUVE points over VP-BART. Blueprint models are also considerably better at grounding and reducing intra- and inter-sentence repetition. The iterative blueprint model obtains a faithfulness score of 51.66% which suggests that more than half of the QA pairs are effectively incorporated into the final story. All models fall short of the upper bound. We should also note that automatic evaluation results are complemented with human evaluation (see Table 3, third block) which confirm that (iterative) blueprint stories are perceived as overall significantly better than VP-BART.
>
> Using gold blueprints yields a superior N-gram score. This is largely due to the overlapping N-grams in the gold blueprints. However, as discussed in Section 4.4, lexical matching metrics like these might not always align with human judgments and may not be the most representative metric for story quality.
>
> Our blueprint method allows for tailored adjustments and controllable generation. In Appendix A.1, we show how blueprint quality can be improved removing hallucinated QA pairs, further refining the story generation process.
>
> > Q2: The current manuscript does NOT compare the proposed method with some strong and pioneering studies [1,2] in the experiments. In my current understanding, the authors exclude them because the N-gram-based performance of the proposed method lags behind that of those studies. It would be much more convincing if the authors compare the proposed method with them. The source code of both studies is open-sourced.
>
> Response: Our decision to exclude these studies [1,2] was primarily due to the page limit, and not an oversight. We prioritized presenting comparisons with the latest state-of-the-art models, such as PR-VIST and KG-Story. It's worth noting that both PR-VIST and KG-Story have been benchmarked against AREL and/or GLAC in the literature, and found to outperform them.
>
> However, in light of your comments and the importance of a comprehensive comparison, we will compare our method with these studies [1,2]. Here are the updated results:
>
> |Model           |Rep-Intra|Rep-Inter|Ground-P.|Ground-R.|MAUVE |B4  |RLSum|
> |----------------|---------|---------|---------|---------|------|----|-----|
> |GLAC Net        |2.94     |84.72    |3.5      |2.1      |1.90  |10.7|33.7 |
> |AREL            |11.96    |84.91    |4.2      |3.3      |2.32  |**14.4**|**35.4**|
> |Ours (top-down) |**0.08**     |81.51    |5.17     |**11.56**    |8.32  |9.9 |33.6 |
> |Ours (iterative)|0.29     |**72.70**    |**5.22**     |3.59     |**28.25** |7.0 |30.3 |

---

### Official Review · Reviewer_1NBh · 2023-08-05

**Soundness:** 3

**Excitement:**

3: Ambivalent: It has merits (e.g., it reports state-of-the-art results, the idea is nice), but there are key weaknesses (e.g., it describes incremental work), and it can significantly benefit from another round of revision. However, I won't object to accepting it if my co-reviewers champion it.

**Paper Topic And Main Contributions:**

This paper introduces planning into visual storytelling. In their case, the plans are a list of QA pairs, which are called blueprints. The method first translates images into visual prefixes based on an image encoder, then iteratively feed the visual prefixes along with the "blueprint" to iteratively generate stories. The authors conduct automatic and human evaluation on the VIST benchmark and show their method achieves good results.

**Reasons To Accept:**

1. The author brings a new iterative, plan-based method into visual storytelling.
2. The results on both automatic and human evaluations are very good.

**Reasons To Reject:**

The main concern for me on this paper is the way they augment the VIST dataset with language models to generate QA pairs as the blueprint plan. In section 3.1, the blueprints B_i are generated by the stories S_i. And then the task is to generate stories S_i based on images I_i and blueprints B_i. This makes the proposed task to have potential data leaking. Say, to the extreme, if the blueprint generator is an identical function, the blueprints will just be the stories. We can also see in Figure 3. that the blueprints and stories have a large number of N-gram overlap. This makes me wonder if the method is ill-defined.

**Reproducibility:**

3: Could reproduce the results with some difficulty. The settings of parameters are underspecified or subjectively determined; the training/evaluation data are not widely available.

**Reviewer Confidence:**

3: Pretty sure, but there's a chance I missed something. Although I have a good feel for this area in general, I did not carefully check the paper's details, e.g., the math, experimental design, or novelty.

---

> ### Author Rebuttal · Authors · 2023-08-28
>
> > The main concern for me on this paper is the way they augment the VIST dataset with language models to generate QA pairs as the blueprint plan. In section 3.1, the blueprints B_i are generated by the stories S_i. And then the task is to generate stories S_i based on images I_i and blueprints B_i. This makes the proposed task to have potential data leaking. Say, to the extreme, if the blueprint generator is an identical function, the blueprints will just be the stories. We can also see in Figure 3. that the blueprints and stories have a large number of N-gram overlap. This makes me wonder if the method is ill-defined.
>
> Response: Thank you for your comments! The blueprints are only seen during training, they are **predicted** during inference, please see more explanation below:
>
> **Fine-tuning Phase**: During fine-tuning, our model takes the image sequence as input and is trained to generate the blueprint **and** the story. The loss is computed based on a comparison between the predicted blueprint+story and the corresponding gold standard blueprint+story.  Fine-tuning is performed on **the training split** of the VIST dataset. The model has no access to the test data during this phase.
>
> **Inference Phase**: During inference the model only sees the image sequence. It does not have access to any gold standard information, For a more detailed discussion please see the Introduction and Section 3.2 of our paper. Figure 3, shows how our model words during inference. The blueprints presented here are **model-generated** and constitute a **plan** of how the story should be realized.
>
> **N-gram Overlap**: The N-gram overlap between blueprints and stories is intentional. The blueprint’s QA pairs are designed to reflect salient content (rendered in natural language) and derived from the stories. This alignment simplifies the generation process and embodies the essence of our planning-based approach.
>
> We hope that this clarification adequately addresses your concerns.
>
> [Note that our gold-standard blueprints are technically silver-standard, see response to reviewer 1. Here we used a simplified terminology to avoid further confusion.]

---

### Official Review · Reviewer_9ntA · 2023-08-11

**Soundness:** 3

**Excitement:**

3: Ambivalent: It has merits (e.g., it reports state-of-the-art results, the idea is nice), but there are key weaknesses (e.g., it describes incremental work), and it can significantly benefit from another round of revision. However, I won't object to accepting it if my co-reviewers champion it.

**Paper Topic And Main Contributions:**

This work introduces a new approach for generating stories from a set of images, task also know as visual storytelling. The authors propose the usage of planning in the form of (question, answer) pairs to guide the story generation and generate more natural and grounded stories.

Briefly, the visual inputs are processed by a frozen image encoder and concept embedder. Output of those is further passed through feed-forward layers and an embedding layer respectively, and finally concatenated to build the input for a sequence-to-sequence BART model. Next, the pre-trained BART model generates the blueprint (question, answer pairs) and the story. This is either done in one go (top-down approach) or iteratively (as questions and story sentences are generated at each step and become input for the next steps).

Question-answer pairs are generated automatically using pre-trained SOTA question generation models. Answers, that are given as input together with the ground truth story are automatically extracted from the story as the noun and verb phrases of each sentence.


**Questions For The Authors:**

Is there any reason you did not include RoVIST-VG[1] metric in your results section?
Also, a short discussion about if and how the proposed architecture can deal with temporal misalignment for this task (e.g. when a concept appears in the visual feed in the third image but in the ground truth story in the first sentence).

In Table 3 (human evaluation) the model with planning and iterative generation is kind of close in performance to VP-BART (same model without planning). In table 2 the VP-BART + BP seems a lot better (e.g. MAUVE metric). Why do you believe this is the case? On metrics one seems much better, while for human evaluation they are very close.


[1] Eileen Wang, Caren Han, and Josiah Poon. 2022. Rovist: Learning robust metrics for visual storytelling. In Findings of the Association for Computational Linguistics: NAACL 2022, pages 2691–2702.


**Reasons To Accept:**

This work introduces planning (in the form of question-answer pairs) for the task of visual storytelling. The idea is well explained, and it works! The contributions and the setup are clear, the results are good and well discussed.

**Reasons To Reject:**

I believe a more thorough analysis of the answer extraction and question generation is needed. As the “ground truth” blueprints are obtained using a learned model (T5) it would be interesting to see how the quality of the question impacts the generated story.

Also there should be clear steps towards preventing data leaking when building blueprints, as they are built from stories themselves.

**Reproducibility:**

4: Could mostly reproduce the results, but there may be some variation because of sample variance or minor variations in their interpretation of the protocol or method.

**Reviewer Confidence:**

3: Pretty sure, but there's a chance I missed something. Although I have a good feel for this area in general, I did not carefully check the paper's details, e.g., the math, experimental design, or novelty.

**Typos Grammar Style And Presentation Improvements:**

-	L939: Table 6 -> Figure 6
-	In my opinion, results of VP-BART with “gt” blueprints should be in a separate table. It makes Table 2 hard to follow.
-	Please provide more details (maybe a footnote) about the concept detector used. Is this model used? https://clarifai.com/clarifai/main/models/general-image-detection?tab=overview .

---

> ### Author Rebuttal · Authors · 2023-08-28
>
> > Q1: I believe a more thorough analysis of the answer extraction and question generation is needed. As the “ground truth” blueprints are obtained using a learned model (T5) it would be interesting to see how the quality of the question impacts the generated story.
>
> Response: Our work uses "silver standard" blueprints. The latter are not manually created but automatically derived from gold standard stories. Specifically, an answer extractor and question generator. Adopting this approach allows us to scale our experiments more effectively. Employing gold standard blueprints would require manual annotation, which is both time-consuming and costly. A recent annotation study using the Question under Discussion theory (which inspired our blueprints) further highlights QUDs are subjective and  different readers often come up with distinct questions even with the same answer sentence (see https://aclanthology.org/2023.findings-acl.710)
>
> > Q2: there should be clear steps towards preventing data leaking when building blueprints, as they are built from stories themselves.
>
> Response: Please see response to Reviewer 2.
>
> > Q3: Is there any reason you did not include RoVIST-VG metric in your results section?
>
> Response: Thank you for bringing this up! We left out the RoVIST metrics because they are very similar to the metrics we used in our study. Specifically, the three components of RoVIST—Coherence, Non-redundancy, and Visual Grounding—match closely with our chosen metrics: the MAUVE score, intra/inter story repetition, and grounding precision/recall.
> Therefore, we focused on these more common metrics in the manuscript.
>
> However, it is straightforward to add these RoVIST results. Due to a suspected bug in the official Visual Grounding script, we couldn't generate results for that component. Thus, we present only the Coherence and Non-redundancy scores below:
>
> |model|non-redundancy|coherence|
> |-|-|-|
> |MCSM|0.90|0.67|
> |KE-Story|0.99|0.64|
> |Ours (iterative)|0.93|0.72|
>
> > Q4: Also, a short discussion about if and how the proposed architecture can deal with temporal misalignment for this task (e.g. when a concept appears in the visual feed in the third image but in the ground truth story in the first sentence).
>
> Response: Thank you for bringing this up! If a visual concept in the third image gets mentioned in the story's first sentence, it suggests that the human annotator felt this was a good way to make the story flow better, even if that concept wasn't present in the first image. They might prioritize narrative flow over strict temporal order, human storytelling often relies on imaginative sequencing.
>
> As the blueprints offer a way to control the generation process (see Appendix A.1, A,2 for examples of how the output can be controlled), it is possible to eliminate misalignment, e.g., by filtering out QA pairs whose entities are not grounded in the corresponding image. This would be implemented using a pretrained language-vision model like GLIP (​​https://github.com/microsoft/GLIP).
>
> > Q5: In Table 3 (human evaluation) the model with planning and iterative generation is kind of close in performance to VP-BART (same model without planning). In table 2 the VP-BART + BP seems a lot better (e.g. MAUVE metric). Why do you believe this is the case? On metrics one seems much better, while for human evaluation they are very close.
>
> Response: In the human evaluation (Table 3), the scores for fluency and coherence are indeed close between the two models. However, for attributes like interestingness and grounding, the model with blueprints stands out as noticeably better than VP-BART without planning. These attributes, which play a key role in assessing story quality, demonstrate that the model with blueprints performs significantly better than the VP-BART without planning.
>
> This aligns with the findings in Table 2. The VP-BART model scores higher on lexical matching metrics, reflecting its strength in fluency and coherence. Yet, its lower MAUVE and grounding scores show that the stories created using the blueprint model feel more real, interesting, and well-anchored in the visual context.

---

### Meta-Review · Area_Chair_53Qb · 2023-09-21

**Recommendation:** 3

**Metareview:**

This paper introduces the use of planning in the form of question-answer pairs to guide visual story generation for more natural and grounded stories. On one hand, reviewers found that (1) the idea is interesting and well explained. The setup is clear and results on both automatic and human evaluations are good and well discussed. On the other hand, two reviewers commented that the impact of generating blueprints seems quite marginal as seen in Tables 2 and 3 (especially for fluency and coherence metrics). The authors did not agree on this, and argue that the model achieves better performance in terms of interestingness and grounding than VP-BART.

---

### Decision · Program_Chairs · 2023-10-07

**Decision:**

Accept-Findings

**Comment:**

This paper introduces the use of planning in the form of question-answer pairs to guide visual story generation for more natural and grounded stories. On one hand, reviewers found that (1) the idea is interesting and well explained. The setup is clear and results on both automatic and human evaluations are good and well discussed. On the other hand, two reviewers commented that the impact of generating blueprints seems quite marginal as seen in Tables 2 and 3 (especially for fluency and coherence metrics). The authors did not agree on this, and argue that the model achieves better performance in terms of interestingness and grounding than VP-BART.